# Evaluation of a Semi-Automated Ultrasound Guidance System for Central Vascular Access

**DOI:** 10.3390/bioengineering11121271

**Published:** 2024-12-15

**Authors:** Sofia I. Hernandez Torres, Nicole W. Caldwell, Eric J. Snider

**Affiliations:** Organ Support and Automation Technologies Group, U.S. Army Institute of Surgical Research, Joint Base San Antonio, Fort Sam Houston, San Antonio, TX 78234, USA

**Keywords:** military medicine, vascular access, automation technology, ultrasound imaging, medical device testing, animal testing, usability assessment

## Abstract

Hemorrhage remains a leading cause of death in both military and civilian trauma settings. Oftentimes, the control and treatment of hemorrhage requires central vascular access and well-trained medical personnel. Automated technology is being developed that can lower the skill threshold for life-saving interventions. Here, we conduct independent evaluation testing of one such device, the Vu-Path™ Ultrasound Guidance system, or Vu-Path™. The device was designed to simplify needle insertion using a needle holder that ensures the needle is within the ultrasound field of view during its insertion into tissue, along with guidance lines shown on the user interface. We evaluated the performance of this device in a range of laboratory, animal, and human testing platforms. Overall, the device had a high success rate, achieving an 83% insertion accuracy in live animal testing across both normal and hypotensive blood pressures. Vu-Path™ was faster than manual, ultrasound-guided needle insertion and was nearly 1.5 times quicker for arterial and 2.3 times quicker for venous access. Human usability feedback highlighted that 80% of the participants would use this device for central line placement. Study users noted that the guidance lines and small form factor were useful design features. However, issues were raised regarding the needle insertion angle being too steep, with potential positioning challenges as the needle remains fixed to the ultrasound probe. Regardless, 75% of the participants believed that personnel with any level of clinical background could use the device for central vascular access. Overall, Vu-Path™ performed well across a range of testing situations, and potential design improvements were noted. With adjustments to the device, central vascular access can be made more accessible on battlefields in the future.

## 1. Introduction

Traumatic hemorrhage management is critical, as it remains the leading cause of preventable death on the battlefield and the second leading cause of death in the civilian sector [1]. A key medical intervention for massive blood loss is providing fluids and therapeutics to maintain blood pressure, a task that depends on quick and reliable vascular access [2,3]. During prior United States Armed Forces conflicts, military medical evacuations (MEDEVACs) of combat casualties from the battlefield were swift, occurring within the “golden hour” due to air superiority, allowing for the prompt treatment of hemorrhage-inducing injuries [4]. However, recent data from the Russo-Ukrainian war have shown that future conflicts with peer and near-peer adversaries will produce mass casualties, leading to delayed evacuations of up to 72 h without air superiority [5]. Further, adversaries have intentionally targeted MEDEVAC personnel and medical facilities treating combat casualties [4,6]. Due to these challenges, it will be important to be able to provide hemorrhage mitigation at or near the point of injury.

The treatment of traumatic hemorrhage requires the control of blood loss followed by replacement fluid delivery to achieve a target blood pressure. The control of extremity hemorrhage may initially involve tourniquet placement, which has a high success rate [7,8]. Conversely, truncal hemorrhage that cannot be controlled through traditional circumferential tourniquets may require damage control surgery, which cannot be performed in the early echelons of care [9]. As such, the placement of a resuscitative endovascular balloon occlusion of the aorta (REBOA) catheter may provide hemorrhage control at the aortic flow level until surgical intervention is feasible [10,11]. Once hemorrhage control is reached, resuscitation with whole blood and other products becomes the primary treatment approach. While peripheral and osseous access devices allow for fluid delivery, central venous access is the only feasible option when high flow rates are needed, as is often the case for trauma-induced hemorrhage [2]. Thus, for REBOA catheter placement and high-volume fluid resuscitation, access to central vasculature is required.

Typically, central vascular access is obtained in a medical facility by a trained medical provider, such as a physician, physician assistant, or nurse practitioner. It is usually guided with ultrasound (US) to make sure that the correct vessel is being targeted and placement is accurate [12]. To be proficient, providers receive extensive training and hands-on practice. Since timely MEDEVAC will not always be possible during future military conflicts, critical interventions like central line insertion will need to be delivered at or near the point of injury, rather than in hospitals where trained medical providers are readily available [13].

As such, there is an urgent need for automated central vascular access devices (ACVADs) that allow non-expert personnel to perform this life-saving measure efficiently and effectively. The technology must provide an easy-to-follow interface to ensure that the procedure can be performed smoothly and without compromising patient safety. Autonomous or semi-autonomous devices would not only help soldiers on the battlefield but could be a force multiplier in military and civilian treatment facilities by disburdening medical providers and allowing additional personnel to perform vascular access procedures, potentially making cannulation quicker, compared to the traditional technique.

Our team performed an independent evaluation of a semi-automated vascular access device developed by Crystalline Medical, Inc (Fremont, CA, USA), called the Vu-Path™ Ultrasound Guidance System. Vu-Path™ is a portable, FDA-cleared US device equipped with a probe and attached needle guide to keep the needle in view of the US. The US display provides track lines to delineate the expected insertion path. Our team previously developed laboratory test platforms [14] for vascular access device evaluation as well as an ex vivo swine model [15]. We expanded on the testing pipeline to evaluate the Vu-Path™ ACVAD in a swine model as well as conduct a human usability assessment. In this effort, we evaluated the technology across this testing pipeline to fully characterize the potential of the ACVAD to simplify the vascular access procedure in future emergency medicine and combat casualty care situations.

## 2. Materials and Methods

### 2.1. Overview of the Vu-Path^TM^ Ultrasound Guidance System

The Vu-Path™ Ultrasound Guidance system is a semi-automated central vascular access device (ACVAD) that incorporates a needle guidance system with a pen-sized ultrasound, as shown on the diagram in Figure 1. The needle is in line with the ultrasound transducer, resulting in the tip remaining in view throughout needle insertion. Custom ultrasound software provides a yellow guideline (Figure 1C) for precisely delineating the needle insertion path. The device was developed to work in a portable laptop format with four major components: (i) ultrasound probe, (ii) needle guide attachment, (iii) US engine, and (iv) a laptop with Vu-Path™ proprietary software. The pen-sized US is a phased-array, broadband transducer with a center frequency of 10MHz, with a focal depth of 2.5 cm and max depth of 4.5 cm. The device allows for sub-millimeter axial resolution and millimeter lateral resolution. The US software enables users to manipulate most of the commonly used US image settings, such as frequency, depth, gain, and color Doppler. The needle guide allows smooth motion of the needle while maintaining friction to minimize needle jostling. A small tab on the side can be pressed after vessel cannulation to detach the needle from the probe.

### 2.2. Laboratory Testing Platforms

#### 2.2.1. Ultrasound Characterization Testing

Two standardized phantoms were created to assess needle visualization at varying depths as well as visualize and perform cannulation of different diameter channels using the Vu-Path™ device. Testing results for each phantom platform were compared to a traditional clinical US system (Sonosite Edge; Fujifilm, Bothwell, WA, USA) by determining signal-to-noise ratio and visibility of the channel’s lumen. For needle visualization across various depths, a rectangular phantom mold (30.5 cm × 15.2 cm × 30.5 cm) was fabricated from laser-cut acrylic sheets (McMaster-Carr, Elmhurst, IL, USA), pasted with acrylic cement (McMaster-Carr, Elmhurst, IL, USA), and secured with foam tape (McMaster-Carr, Elmhurst, IL, USA). The phantom was made of 15% *w*/*w* gelatin (Thermo-Fisher, Waltham, MA, USA) solution in a 2:1 *v*/*v* water to evaporated milk (Nestle, Vevey, Switzerland) mixture, with 0.50% (*w*/*v*) flour as the scattering agent [16]. A 6.8 mm wire was suspended at 45°, as diagrammed in Figure 2A to simulate a needle at increasing depths and imaged every 1.3 cm (Figure 2B) across the phantom until the deepest US system setting was reached. The second phantom (30.5 cm × 20.3 cm × 7.6 cm) was made using the same fabrication technique and phantom recipe, with seven channels simulating vessels of different diameters: 2.5, 3.2, 4.0, 5.0, 6.1, 8.0, and 9.6 mm (Figure 2C), and poured out of the same gelatin mixture. Vessel centers were 3.8 cm beneath the surface of the phantom and evenly distributed across the 30.5 cm length of the phantom. The channels were imaged with both US systems for vessel lumen identification (Figure 2D), and three insertions were attempted across all channel diameters using an 18-gauge needle (Sensi-Touch, Sherwood Medical, St. Louis, MO, USA) with the ACVAD. Success was determined by flashback, using a syringe attached to the needle.

#### 2.2.2. Commercial Vascular Access Trainer

A commercial tissue phantom for femoral line insertion training (CAE Blue Phantom, Sarasota, FL, USA), was also used to evaluate needle insertion with the ACVAD. The model has multiple channels simulating major vessels in the femoral region and handpumps to enable simulated pulsatile flow in the artery. With the ACVAD, three insertion attempts were allowed at each vessel, timing key steps, starting when the US probe made contact with the phantom surface, when target was identified, and finally when needle was placed. At this point, successful insertion was determined by needle flashback. The commercial phantom had blue fluid in the simulated veins and pink fluid in the arteries. If the insertion was successful, a secondary ultrasound was used to capture the angle of insertion and distance from the vessel center for each attempt.

#### 2.2.3. Custom Tissue Phantom for Vascular Access

As a final phantom test platform, the Vu-Path™ device was evaluated using a custom, modular tissue-mimicking phantom that can easily be adjusted to multiple different vessel diameters representing anatomical properties of different physiological states and species [14]. For this use case, the anatomy chosen was representative of normotensive and hypotensive femoral swine vessels to aid in readying the device for animal testing. The design of the tissue phantom vasculature (Figure 3) can be modified on a case-by-case basis to represent vessels of different diameters, paths, and physiological states. The tissue phantom was made with a 15% gelatin mixture of 2:1 water to evaporated milk containing 0.50% flour, and vessels were simulated with latex tubing (Cole Parmer, Vernon Hills, IL, USA) of the corresponding diameter. The phantom was connected to a flow loop with a pulsatile, heart-mimicking pump (ViVitro Labs, Victoria, BC, Canada) circulating water, and pressure was quantified by sensors connected to a patient monitor (Draeger, Lubeck, Germany), which confirmed physiological status (Figure 3A). Once the tissue phantom was ready and set up within the flow loop, three insertion attempts were performed on each vessel for each physiological state. Then, after successful insertion was confirmed by flashback, US scans were captured with the clinical US system to determine angle of insertion and distance of needle tip from vessel center.

#### 2.2.4. Ex Vivo Swine Vascular Access Model

A previously described lower-body ex vivo porcine model [15] was used to test device performance on swine tissue. Briefly, the lower-body section was perfused through cannulations proximal and distal to the femoral vessel insertion region (Figure 4). The arterial and venous circulation was connected via an arterio-venous shunt, and the model was perfused with a flow loop circulating water with a peristaltic pump (Cole Parmer, Vernon Hills, IL, USA), connected to a patient monitor to ensure pressure regulation. This allowed for simulation of normotensive and hypotensive conditions for needle insertion with the Vu-Path™ device. A single successful attempt per vessel at each physiological state was allowed using this model. The same metrics (flashback, insertion angle, and distance from center) were evaluated for each attempt.

### 2.3. Live Animal Model

All procedures using live swine animals were performed using subjects from two approved research protocols. Research was conducted in compliance with the Animal Welfare Act, the implemented Animal Welfare regulations, and the principles of the Guide for the Care and Use of Laboratory Animals. The Institutional Animal Care and Use Committee at the United States Army Institute of Surgical Research approved all research conducted in this study. The facility where this research was conducted is fully accredited by the AAALAC International. Live animal subjects were maintained under a surgical plane of anesthesia and analgesia throughout the studies. The first animal study evaluated a swine model of acute kidney injury by ischemia reperfusion due to hemorrhagic shock and included a 24 hr ICU follow-up, followed by euthanasia. The needle insertions related to evaluation of the Vu-Path™ device were made after all study events occurred, prior to euthanasia. At this point, the subjects were normovolemic. The second animal study evaluated performance of automated hemorrhagic shock controllers and consisted of two consecutive hemorrhage resuscitation events, followed by euthanasia. After the first hemorrhage resuscitation event, the swine subject was normovolemic/normotensive, and insertions with the ACVAD were attempted into both the femoral artery and vein. Then, after the second hemorrhage to a mean arterial pressure of 35 mmHg, hypovolemic/hypotensive insertions using the ACVAD into both femoral vessels were performed.

Three needle insertion datasets were captured across both animal studies: using the Vu-Path™ device with the needle clip at a (i) normotensive or (ii) hypotensive state; and (iii) manual needle insertion at normotensive state using only the Vu-Path™ ultrasound probe. Each dataset was from *n* = 5 subjects for both venous and arterial insertion. Time measurements were recorded for all insertions described above to quantify time to find target and time to place needle. Testing was completed after successful insertion, as measured by flashback, but a maximum of three attempts were allowed for each vessel, at each state. Evaluation of needle placement was performed with a secondary clinical ultrasound (Sonosite PX, Fujifilm, Bothwell, WA, USA) to capture images for later analysis.

### 2.4. Human Usability Assessment

This research was conducted in compliance with a protocol reviewed and approved by Headquarters U.S. Army Medical Research and Development Command Institutional Review Board. The study recruited participants who met the following four inclusion criteria: (1) individuals 18 or older; (2) individuals working or training at Joint Base San Antonio, Fort Sam Houston; (3) individuals with training or experience in IV/A-line insertion (animal or human); and (4) individuals willing to perform a simulated central line insertion on a synthetic vascular access trainer and provide feedback. Individuals with scheduling conflicts or constraints who could not dedicate time to the research were excluded from participating in this study. Participation in this study was completely voluntary, and participants were not compensated for their time.

For this study, consented participants (*n* = 20) were asked to watch a standardized training video on how to use the Vu-Path™ device and were then instructed to use the device to insert a needle into the femoral vein of the commercial tissue phantom trainer (Section 2.2.2). A maximum of five attempts, each lasting no longer than five minutes, were allowed per participant, and success was confirmed by either venous or arterial fluid flashback. In accordance with the other laboratory testing approach described above, timing was measured from contact with the tissue phantom for the identification of vessels and completion of needle insertion. All attempts were evaluated for needle flashback, and successful placements were captured by a secondary US system.

After phantom testing, the participants were asked to respond to a survey consisting of questions regarding their demographic information and prior experience, as well as binary (“Yes” or “No”) questions regarding Vu-Path™ device training, design, ultrasound, and operability. The final part of the study consisted of discussion questions with open answers, aimed at obtaining more granular feedback about the device in its current form, potential improvements to be made, and where the device could optimally be used.

### 2.5. Analyzing Performance Across the Test Platforms

The calibration results were analyzed using FIJI to determine the signal-to-noise ratio of the needle depth visualization and lumen visibility at varying diameters [17,18]. This was measured through image histogram analysis across a line bisecting the needle or vessel feature in the image to calculate average signal intensity values, as required. FIJI was also used for characterizing needle insertion across animal and phantom models, using two sets of US scans: a cross-sectional view of the needle tip and an in-plane view of the needle in the vessel. These were captured as videos with a secondary, clinical US machine (Sonosite Edge for tissue phantoms and ex vivo model and Sonosite PX for live animal and human usability studies). The cross-sectional captures were used to determine the distance of the needle tip to the vessel center. For the in-plane view, the angle at which the needle was inserted into the vessel was calculated. FIJI was used for analysis of images captured with the Sonosite Edge while built-in calculations for obtaining these measurements were used for images captured with the Sonosite PX. For human usability survey responses, data were summarized regarding clinical experience and demographics. For binary survey responses, data were compiled as percentage of “Yes” or “No” responses. For discussion questions, data were tallied across all participants regarding general response themes on various discussion topics.

### 2.6. Statistical Analysis

All results are presented as mean values with error bars denoting standard deviation unless otherwise indicated. Timing results are shown as time to identify vessel, time to obtain vascular access, and overall time from beginning to end. Overall success rates of vascular access are indicated and differentiate between vein and artery access and normotensive and hypotensive pressure conditions when possible. To minimize the effects of user experience on testing results, all testing was performed by a single experienced user, except for the human usability study.

All analyzed data visualizations were conducted using Prism 10.3 software (GraphPad, La Jolla, CA, USA). In vivo vascular access testing was assessed for statistically significant differences between normotensive and hypotensive access as well as automated and manual access attempts. Normality was evaluated by Shapiro–Wilk test, and datasets were identified as non-normally distributed. As such, differences were assessed by Kruskal–Wallis test with Dunn’s post hoc test to compare differences between groups. *p*-values below 0.05 denote statistically significant differences.

## 3. Results

### 3.1. Performance of the Vu-Path^TM^ Device in Laboratory Models

Testing was performed using ultrasound characterization phantom, tissue test phantoms, and an ex vivo animal model in the initial phase of the testing pipeline. The characterization phantom models aimed to evaluate the ultrasound image obtained from the Vu-Path™ device and determine its feasibility for accessing an artery or vein. The characterization of the US images assessed the signal-to-noise ratio of those obtained from the ACVAD compared to a clinical US when visualizing a target at predetermined depths. The signal-to-noise ratios were comparable for each depth except at approximately five cm, where Vu-Path™’s ratio was 1.7 and that of the clinical US was 3.1 (Figure 5A–C). A different characterization test used different-sized channels to determine the visualization of vessel lumen and determine if insertion was possible using the ACVAD. The results show that for vessel diameters of 6.1 mm or less, the vessel lumen was not distinguishable (Figure 5D–F). However, the overall structure could be identified and still resulted in 100% successful insertions at nearly every diameter size using the ACVAD, except for 3.2 mm.

Next, two different tissue phantoms were used to evaluate the Vu-Path™ device—a commercial and a custom tissue phantom model. Needle insertion was characterized in each, with the commercial phantom resulting in 100% successful insertions (Figure 6A,B) and the custom setup having a success rate of 87% (Figure 6E,F). The time required to identify and cannulate vessels was similar for both artery and vein insertions at 24.5 s vs. 23.9 s, respectively (Figure 6C). The needle insertions for the commercial trainer had an average distance from the needle tip to the vessel center of 1.41 ± 1.29 mm, while the angle of insertion was quite steep at 86.3 ± 3.3° across both arterial and venous features (Figure 6D). The results were similar for the custom phantom setup with an average insertion angle of 81.7 ± 14.6° and distance from center at 1.2 ± 0.82 mm (Figure 6G,H). The custom phantom allowed for evaluating performance differences with vessels of sizes mimicking arteries and veins at hypovolemic and normovolemic levels. The differences were most obvious for normovolemic insertions, which took longer than hypovolemic insertions (27.1 s vs. 17.4 s), as well as venous insertions, which took longer than artery insertions (27.5 s vs. 17.0 s; Figure 6G).

An evaluation of the device under hypotensive and normotensive conditions was also performed in an ex vivo animal model. This model better mimicked physiological needle insertion compared to tissue phantom models prior to live animal testing. Due to the limitations of the model, only one needle insertion was performed in each vessel (femoral artery and vein) at both physiological states. For the normovolemic artery, the first attempt was unsuccessful and thus required a second attempt. All other insertions were successful on the first attempt with an average overall time of 29.2 ± 23.8 s to identify the vessel and an additional 20.6 ± 8.6 s to successfully cannulate. This provided sufficient performance evidence to advance to product testing in a live animal model.

### 3.2. Performance of the Vu-Path^TM^ Device in Live Animal Model

The tests using a swine model were split evenly between normotensive (a MAP above 65 mmHg) and hypotensive (a MAP of approximately 35 mmHg) for both the femoral artery and vein for a total of five attempts for each permutation across seven swine subjects. In addition, five normotensive manual insertion attempts without the needle holder but with the Vu-Path™ device (pen-sized US only) were collected across five swine subjects. Overall, the success rate with the ACVAD was 83% with fewer problems noted in venous (100% success) and normotensive (91% success) access compared to arterial (71% success) and hypotensive (77% success) attempts (Figure 7A). The timing results for successful insertions were comparable, with average results ranging from 25.6 to 22.4 s across all groups (Figure 7B). The distance from the needle tip to the vessel center differed mostly for the hypovolemic artery features, with a distance of 0.88 ± 0.44 mm, compared to the overall performance, a distance of 1.4 ± 0.57 mm (Figure 7C). The distances for normovolemic and hypovolemic artery insertions were found to be statistically significant at 1.8 ± 0.22 mm and 0.88 ± 0.44 mm, respectively (*p* = 0.0125). The insertion angle metrics showed a more acute angle of insertion compared to tissue phantom attempts, with an average insertion at 57.8 ± 13.8° (Figure 7C). The overall insertion success rate was similar during the manual insertion attempt (83.3%), but the timing of the procedure was significantly slower for the vein insertion at 60 ± 16 s vs. 25.6 ± 8.4 s (*p* = 0.014, Figure 7D).

### 3.3. Feedback from Usability Assessment of the Vu-Path^TM^ Device

Through an IRB-approved human use protocol, we collected the user feedback and usability metrics of 20 participants using the commercial tissue trainer to evaluate the ACVAD. The participants were 40 years old on average, having about 8 years of ultrasound experience. All the participants had clinical experience, as summarized in Table 1. The participants were shown a standardized training video prepared by the research team and were then asked to complete at least one successful needle insertion using the test device on the commercial trainer. The time was measured from the first contact of the device with the tissue phantom to successful needle insertion. The target vessel was the femoral vein, which was cannulated by 70% of the participants, while the rest achieved femoral artery access. The average number of insertion attempts was 1.5 with an overall success rate of 64.5%. The timing was slower compared to live animal testing, as this was the participants first time using the device, with the average time for a successful attempt being 197 ± 123 s (Figure 8A). The distance from the center of the vessel to the needle tip was 2.0 ± 0.10 mm, while the insertion angle was 76.5 ± 10.1° (Figure 8B).

After using the device, the study participants completed a survey about their experience and answered open-ended discussion questions. The responses for relevant questions are shown in Table 2. Summarized survey responses for all questions are shown in Appendix A. Overall, 95% of the users found the device easy or safe to use. Similarly, 85% indicated they liked using the device, with 80% indicating they would use Vu-Path™ for central line placement and 85% indicating that the device was suitable for remote, austere environments. Over half of the participants (65%) thought the device could be used by someone without central line placement experience, while 45% reported challenges orienting the ultrasound probe. Four users (20%) reported frustration when using the device, but thirteen participants (65%) found identifying the vessel target using the device easy. From the open discussion, 40% (*n* = 8) of the participants indicated the on-screen track lines were a useful feature, while others indicated that the means by which the needle was held and the portability of the system were useful design aspects (Figure 8C). Conversely, 40% (*n* = 8) of the users indicated that the steep angle of insertion was an issue with the device for this vascular access application, and others found the needle and ultrasound being fixed together cumbersome at times (Figure 8D). The most common recommended changes to the device were in regards to ergonomics, improved needle detachment, ultrasound orientation assistance, and changes to needle positioning (Figure 8E). Even with these issues raised, 75% (*n* = 15) of the participants thought any-level medical provider could use this device, 10% (*n* = 2) thought the device was best suited for mid-level medical providers and above, and 10% (*n* = 2) thought mid-level medical providers and below were the most suitable use case for this device (Figure 8F).

## 4. Discussion

Following trauma, medical interventions and therapies often require central vascular access for obtaining hemorrhage control, taking reliable blood pressure readings, and administering fluids and medications. Unfortunately, there is a high skill threshold for safe and efficacious central vascular access, something that is typically not feasible in pre-hospital and combat environments. Engineering technology solutions can fill this gap, if properly designed and characterized for this application. In this capacity, ACVADs have the potential to lower the skill threshold by providing appropriate guidance and simplified ultrasound operation for central vascular access.

Overall, the Vu-Path™ device performed well for this vascular access task across the standardized test pipeline we presented, but some performance limitations were identified. The initial evaluation with laboratory models highlighted that the ultrasound transducer could identify deeper vessels and small-diameter features as needed for central hypovolemic vascular access. Vascular access attempts in these laboratory models were quick and highly accurate, justifying the further evaluation of the product in elaborate, physiologically relevant models. Next, the ex vivo swine model results continued to highlight the device’s strong performance in achieving vascular access, but the average times to achieve access were slower due to the higher complexity of underlying tissue being accessed. More extensive evaluations for both hypovolemic and normovolemic situations were assessed by a live animal study. The evaluated ACVAD had an overall 83% success rate in achieving central vascular access across all conditions. There was a higher failure rate with arterial features and hypovolemic conditions compared to venous features and normovolemic access. In a low-blood-volume situation, the small diameter of the artery resulted in a higher failure rate, likely due to the device’s precision and track line feature being similar in size to the small vessel. Through human use testing, the Vu-Path™ device had a higher failure rate and slower average access time than previous evaluations. This was to be expected, as this was the very first time for all the participants using the device, while other validation studies were performed by a single experienced user. Regardless, 80% of the users expressed that they would use the device for vascular access in their current line of work, highlighting the potential utility of the technology.

The testing pipeline did identify some limitations with the current ACVAD technology. First, the insertion angle relative to the vessel wall was steep, often larger than 45° and sometimes approaching 90°. For needle insertion, this is acceptable, but it may lead to issues during guidewire advancement prior to catheter placement. At a perpendicular angle, the wire may get caught against the vessel wall or travel the wrong direction in the artery or vein. Some of this was related to user inexperience as evidenced by the drastic average angle of 76° in human testing, versus 55° in the live animal study in which a single user performed repeated tests using the device. The parallel attachment of the probe and needle could be counterintuitive to some users and different from how ultrasound-guided vascular access is traditionally instructed. Automation or guidance features integrated with the Vu-Path™ device can potentially reduce this concern if properly designed. A second identified issue during human use testing was related to the design of the ultrasound transducer and fixed positioning of the needle holder. The small, pen-like transducer led to confusion as to the orientation and, thus, some users had issues knowing which way to move the probe to center the vessel features in view. The needle holder was most comfortable when the users positioned it above the transducer as opposed to beneath, as is more common for traditional vascular access, which likely influences the steep angle of insertion. A final limitation identified by the study participants was the skill threshold necessary to properly use the device; 75% felt any skill level could use the device. While many believed that a combat medic or similar minimally trained clinician could utilize the device with proper training, many others identified that strong knowledge in ultrasound imaging was still needed, as much of the motion and vessel identification were still manual when using the device. Improving automation or guidance features to provide overlays of vessel identification or assistance with how to move the probe could alleviate these concerns in future device designs.

A secondary achievement of this work is the development of a testing pipeline for the thorough evaluation of vascular access technology. The series of methods start with low-cost, higher-throughput laboratory models that can provide initial troubleshooting or product evaluation. We envisioned the platform with performance gates for each step of the pipeline; until its performance reaches a consistent high success rate, testing methods with fewer regulatory hurdles and less time-consuming experimentation can be withheld. However, these later experiment steps are critical to fully evaluating a product, as laboratory models cannot fully replace live animal testing or human usability assessments. Yet, there were some limitations with the test platform. First, for the laboratory and animal experiments, we were focused on the performance of the vascular access device rather than the user, so only one trained operator performed all the tests. This led to more consistent testing, but there is inherent bias in operator comfortability and preference built into these test platforms. The human usability assessment does mitigate this risk by highlighting the device performance across a wider audience. Second, earlier test platforms used tissue phantom, post-mortem tissue, or commercial trainers, and, as such, there was reduced physiological relevance across some of the testing models, which could impact testing performance results prior to moving to the more realistic testing situations. In fact, a large number of participants in the human use study identified issues with the commercial trainer not being a realistic testing platform, and it could have impacted performance during testing as well. Improved vascular access trainers are still needed and could be easily integrated in the testing pipeline, if identified or developed.

The next steps for this work can be divided into improvements to the Vu-Path™ vascular access product and the testing pipeline. There was an identified need across much of the characterization testing that more automation or guidance features can improve performance and reduce the device skill threshold, which are critical in combat casualty care or pre-hospital conditions. Guidance features for tracking artery and vein locations will reduce the ultrasound skill threshold needed for the current device. Further, needle tracking features to identify when the needle is in the vessel as well as screen prompts for how to manipulate the probe to improve the angle of needle insertion are desirable. For the testing pipeline, improvements can be made to the laboratory models to find better, more realistic tissue trainers to improve early product evaluations. In addition, improvements will be made by expanding product evaluations into human cadaver models so that the performance of the device can be tested on human vessel anatomy to accelerate the product’s translation into a clinical setting. Furthermore, a randomized head-to-head clinical comparison study between the ACVAD product and manual vascular access procedures will be needed to validate performance in future studies.

## 5. Conclusions

Central vascular access is critical for trauma care in the hospital setting as well as remote pre-hospital settings such as on battlefields in the future, where field care may be prolonged for up to 72 h. In these cases and in mass-casualty environments in which there are limited medical personnel, there remains a need to automate or simplify central vascular access. The Vu-Path™ vascular access device characterized in this work has potential to address this issue based on its strong performance across a wide range of testing models. Vascular access was obtained at quicker rates compared to manual ultrasound-guided insertion and had a high success rate for both normovolemic and hypovolemic physiologies. In a human usability assessment, end-users were successful at using the device and highlighted the low-skill threshold for using the device, which is critical in a field setting with limited trained personnel. As such, the technology remains a viable solution for central vascular access, with potential to lower the skill threshold needed to care for patients with hemorrhages.

## Figures and Tables

**Figure 1 bioengineering-11-01271-f001:**
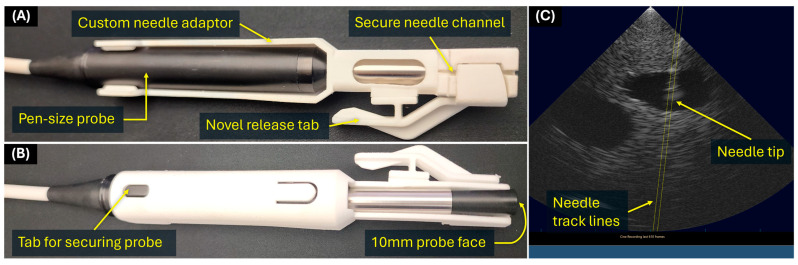
**Annotated Diagram of Vu-Path™ Ultrasound Guidance System.** (**A**) Front of device with labeled features. (**B**) Back of device with labeled features. (**C**) User interface for the ultrasound with track lines (yellow dashes) illustrating needle path and visible needle tip. Lines shown on example ultrasound image were overlayed to enhance figure clarity. An overview video of the device is available at Vu-Path™ website: https://crystallinemed.com (Access Date: 9 December 2024).

**Figure 2 bioengineering-11-01271-f002:**
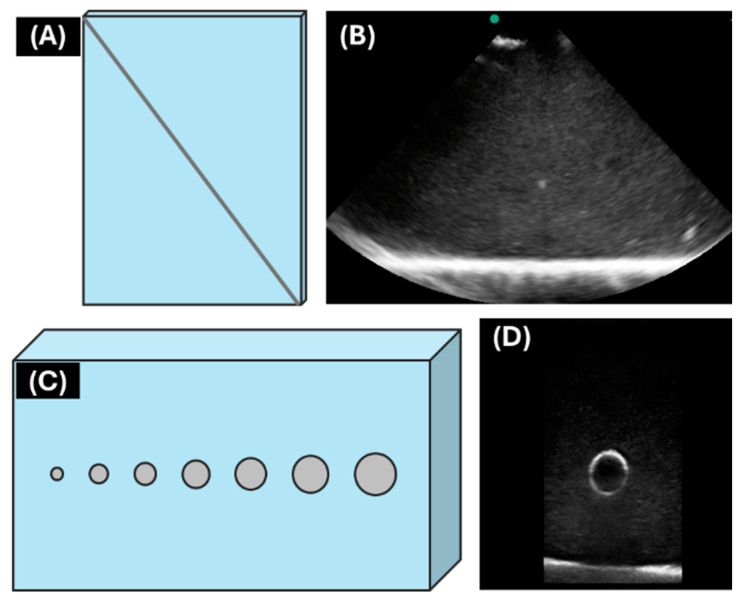
**Drawings of ultrasound characterization phantoms and representative ultrasound images.** (**A**) Diagram of phantom for needle depth assessment with (**B**) representative ultrasound image. (**C**) Diagram of phantom with varying vessel diameters and (**D**) representative ultrasound image. US images shown are from the clinical US system.

**Figure 3 bioengineering-11-01271-f003:**
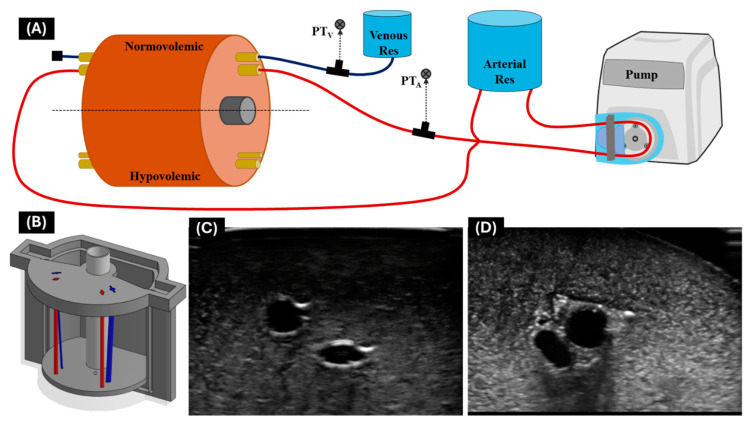
**Modular tissue phantom features.** (**A**) Diagram of flow loop for modular tissue phantom with parts identified. (**B**) Engineering drawing of tissue phantom mold with swine anatomy for hypovolemic and normovolemic vessels. For each, arterial vessels are shown in red while venous vessels are shown in blue. Representative ultrasound images captured for modular tissue phantom for (**C**) normovolemic vessels only and (**D**) normovolemic vessels with a simulated nerve bundle. Reproduced from [14].

**Figure 4 bioengineering-11-01271-f004:**
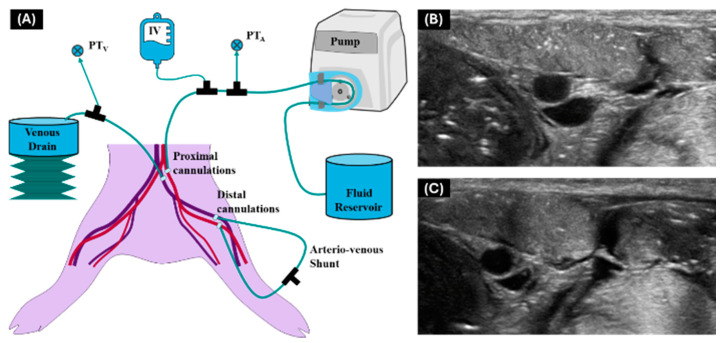
**Setup for ex vivo model with representative ultrasound images.** (**A**) Diagram of flow loop with cannulated vessels for ex vivo porcine model. Arterial vessels are shown in red while venous vessels are shown in blue. Representative ultrasound images at (**B**) normovolemic and (**C**) hypovolemic states. Reproduced from [15].

**Figure 5 bioengineering-11-01271-f005:**
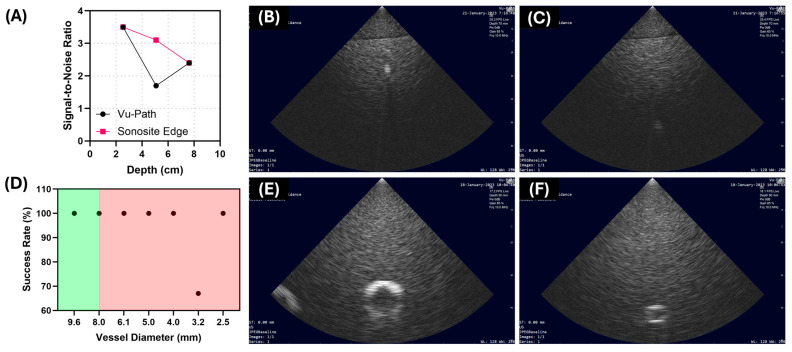
**Summary of results for the Vu-Path™ device in the benchtop characterization phantoms.** (**A**) Summary of performance for the ultrasound depth characterization phantom compared against a clinical ultrasound machine. (**B**,**C**) Representative US images at two different depths with the Vu-Path™ device. (**D**) Summary of performance with vessel diameter characterization phantom. Background color corresponds to vessel lumen visibility status by the device, with green representing channel lumen was visible and red indicating channel lumen was not visible. (**E**,**F**) Representative US images at two vessel diameters imaged with the ACVAD.

**Figure 6 bioengineering-11-01271-f006:**
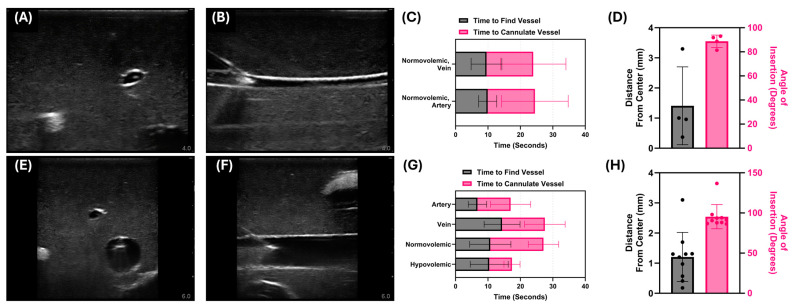
**Summary of Vu-Path™ performance in tissue phantom models.** (**A**,**B**) Representative US pictures of needle insertion in the commercial trainer taken from a cross-sectional and longitudinal view. (**C**) Insertion timing and (**D**) distance and insertion angle performance metrics for the commercial phantom (*n* = 3 attempts). (**E**,**F**) Representative US pictures of needle insertion in the custom phantom trainer taken from a cross-sectional and longitudinal view. (**G**) Insertion timing and (**H**) distance and insertion angle performance metrics for the custom phantom trainer (*n* = 10 total attempts).

**Figure 7 bioengineering-11-01271-f007:**
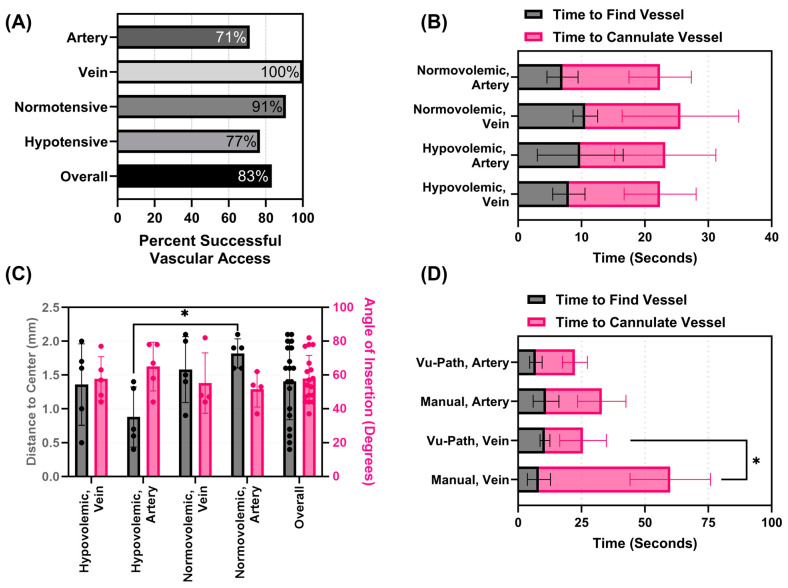
**Summary of results for live animal testing of the Vu-Path™ device.** (**A**) Success rates for live animal studies for arteries and veins, as well as for normotensive and hypotensive vascular access with the ACVAD. (**B**) Summary of needle insertion times for each vessel and physiological condition pairs. (**C**) Distance of needle to vessel center and insertion angle with the ACVAD. (**D**) Comparison of needle insertion timings using the Vu-Path™ device and without (labeled manual) the needle guide clip. Statistical significance (*p* < 0.05) is denoted by asterisk when applicable.

**Figure 8 bioengineering-11-01271-f008:**
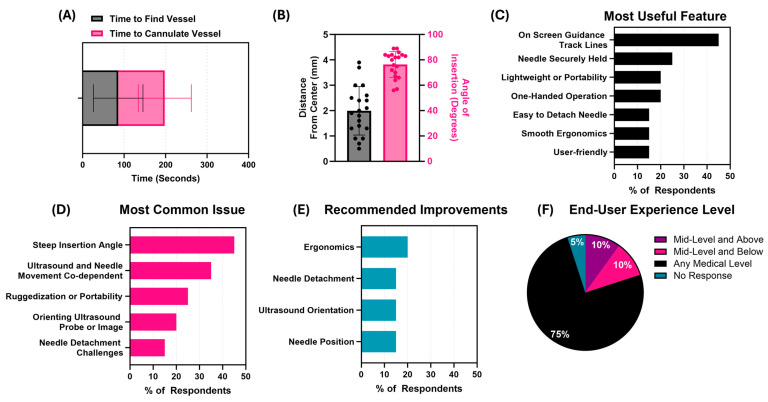
**Summary of Vu-Path™ performance and discussion results from usability study.** (**A**) Average time to find vessel and perform successful cannulation (*n* = 20 participants). (**B**) Distance and insertion angles for all successful insertion attempts. (**C**–**F**) Consensus answers from discussion topics related to (**C**) most useful feature of the device, (**D**) most common issue with the device, (**E**) recommended improvements, and (**F**) recommended experience level for those using the device. Results for (**C**–**E**) highlight opinions from at least 15% of the participants, or three out of twenty of the participants.

**Table 1 bioengineering-11-01271-t001:** **Summary of participant demographics from survey responses.** Data are shown as average with standard deviation for numeric responses and as percentages for categorical ones.

Demographic Category	Distribution
Enrolled participants	20
Average number of attempts for successful insertion in tissue phantom	1.5
Age	40.65 (±8.4) years
Ethnicity	85% non-Hispanic 15% Hispanic
Race	95% White 5% Black, African American
Dominant hand	90% Right 10% Left
Gender	55% Male 45% Female
Clinician type	40% MD/DO 15% RN/LVN/PM 15% Tech/MA 10% PhD, Research Scientist 10% Medic, EMT 5% DVM/VMD 5% NP/PA
Number of central lines previously placed on patients or live animals	8.2 (±13.9)
Number of central lines previously placed using a simulator model	0.9 (±1.9)
Years of experience using ultrasound	8.2 (±7.5)

**Table 2 bioengineering-11-01271-t002:** **Summary of survey responses from human usability study.** Responses are shown as percentage “Yes” response for each.

Question	Percent “Yes” Response
Did you have difficulty orienting yourself to the ultrasound?	45%
Was your insertion target easy to identify on the ultrasound?	65%
Was the device easy to use?	95%
Did you feel safe using this device?	95%
Did you find use of the device frustrating?	20%
Do you think this device could be used by someone without central line placement experience?	65%
Overall, did you like using the device?	85%
Would you use this device to obtain central line access on a patient?	80%
Is this device feasible for use in remote, austere environments?	85%

## Data Availability

The datasets presented in this article are not readily available because they have been collected and maintained in a government-controlled database that is located at the U.S. Army Institute of Surgical Research. As such, data can be made available through the development of a business agreement with the corresponding author. Requests for the datasets should be directed to Dr. Eric J. Snider at eric.j.snider3.civ@health.mil.

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
