# Peer review of "Evaluation of a Semi-Automated Ultrasound Guidance System for Central Vascular Access"

_bioengineering, 2024, doi:10.3390/bioengineering11121271_

Round 1
Reviewer 1 Report
Comments and Suggestions for Authors
The manuscript evaluates the performance of the Vu-Path™ Ultrasound Guidance System for central vascular access. It uses laboratory, animal, and human models to assess accuracy, speed, and usability, and discusses potential improvements. Success rates and user feedback underscore the system's usability, though challenges like needle insertion angles and orientation remain.
Strengths
The manuscript presents a thorough and well-structured evaluation of a new system for vascular access, with clear data supporting its potential to enhance vascular access in critical settings. The inclusion of diverse testing platforms, including human usability studies, adds robustness to the findings. Figures and tables effectively complement the analysis.
Main Critiques
The primary limitation of the system seems to be the steep insertion angle, which could pose significant challenges during guidewire placement and catheter advancement. This issue, noted across multiple tests, needs detailed consideration. The discussion acknowledges it but lacks specific recommendations for resolving the problem. Additionally, reliance on a single experienced operator for many tests limits the generalizability of results. I personally don’t like the repetitive use of the product’s name and would suggest a more general term in text. I also think that a randomized head to head comparison should be planned in the future (and mentioned in the discussion section)
Minor Points of Critique
Several sections are overly technical, potentially hindering accessibility for a broader audience. Simplifying these areas would enhance readability.
The manuscript inconsistently refers to the target vessel, alternately mentioning vein and artery without clarification
While figures are useful, some labels and legends are difficult to interpret at first glance. Improving clarity would help. A video as supplementary material could help to quickly understand the system
General Comments
This is an interesting manuscript, but there are minor areas where clarity can be improved. While the manuscript contributes valuably, it feels repetetive in parts. Sentences like "the device was fast in venous and arterial access" are stated multiple times without new insights. Reducing this repetiton would make the text more concise and impactful.
Author Response
The manuscript evaluates the performance of the Vu-Path™ Ultrasound Guidance System for central vascular access. It uses laboratory, animal, and human models to assess accuracy, speed, and usability, and discusses potential improvements. Success rates and user feedback underscore the system's usability, though challenges like needle insertion angles and orientation remain.
Strengths
The manuscript presents a thorough and well-structured evaluation of a new system for vascular access, with clear data supporting its potential to enhance vascular access in critical settings. The inclusion of diverse testing platforms, including human usability studies, adds robustness to the findings. Figures and tables effectively complement the analysis.
Main Critiques
The primary limitation of the system seems to be the steep insertion angle, which could pose significant challenges during guidewire placement and catheter advancement. This issue, noted across multiple tests, needs detailed consideration. The discussion acknowledges it but lacks specific recommendations for resolving the problem.
We appreciate the reviewer taking the time to read our manuscript. As for the lack of recommendations related to improving the angle of insertion, we identify the challenge but are hesitant to provide recommendations for adjusting the design. We are conducting independent testing separate from the company so providing feedback in this manner would be conjecture that may not be in line with the company’s plans ahead. We have clarified this testing approach as independent testing throughout the manuscript.
Additionally, reliance on a single experienced operator for many tests limits the generalizability of results.
This point is accurate but the experimental tests were trying to test the device performance vs. performance across operators. However, the usability study where 20 different participants tested the device addresses this limitation to see the performance difference across a number of experience levels. This limitation is mentioned in the discussion section.
I personally don’t like the repetitive use of the product’s name and would suggest a more general term in text.
Great suggestion, we have simply referred to the device as ACVAD or automated central vascular access device more routinely through the manuscript instead of the specific product name throughout.
I also think that a randomized head to head comparison should be planned in the future (and mentioned in the discussion section)
We have added reference to need for this type of testing as a next step in the limitations section of the discussion.
Minor Points of Critique
- Several sections are overly technical, potentially hindering accessibility for a broader audience. Simplifying these areas would enhance readability.
- The manuscript inconsistently refers to the target vessel, alternately mentioning vein and artery without clarification
- While figures are useful, some labels and legends are difficult to interpret at first glance. Improving clarity would help. A video as supplementary material could help to quickly understand the system
We have tried to clean up text throughout the manuscript to remove unnecessary technical details and refer to vessels more specifically when possible. Further, figure captions were simplified when possible. A video was not available of the product performance, but we added a reference link to the company’s website where a video is available. It can be found in Figure 1 caption and at the following link: https://crystallinemed.com/.
Reviewer 2 Report
Comments and Suggestions for Authors
This paper evaluates a semi-automated device to assist in need leinsertion for central vascular access. The authors conducted performance tests of the equipment, ex vivo tissues, animal experiments and human experiments, and the entire study design and protocol were comprehensive and detailed. According to the results, the device is helpful in improving the insertion accuracy, and the research is also very meaningful, confirming its usefulness and reliability.
The issues to be aware of are as follows:
1. In Figure 1, Fig.1(c) has track lines, but the image of the needle is not visible, so it is recommended to replace it.
2. Fig.2 There is a title error, the second (c) should be (d). At the same time, it is recommended that it is best to have pictures of blood vessels of different diameters presented side by side.
3. In addition, the detailed performance parameters of the system should be introduced, such as the type, frequency, and number of elements of the ultrasonic transducer used; The imaging resolution, field of view, depth, etc. of the system.
Author Response
This paper evaluates a semi-automated device to assist in needle insertion for central vascular access. The authors conducted performance tests of the equipment, ex vivo tissues, animal experiments and human experiments, and the entire study design and protocol were comprehensive and detailed. According to the results, the device is helpful in improving the insertion accuracy, and the research is also very meaningful, confirming its usefulness and reliability.
The issues to be aware of are as follows:
- In Figure 1, Fig.1(c) has track lines, but the image of the needle is not visible, so it is recommended to replace it.
Thanks for taking the time to evaluate our manuscript. We agree with the reviewer and have added a different image where the needle is more readily identifiable in the vessel.
- Fig.2 There is a title error, the second (c) should be (d). At the same time, it is recommended that it is best to have pictures of blood vessels of different diameters presented side by side.
The title letters have been corrected. Figure 4 in the results shows different vessel diameters to highlight how they vary in size side by side.
- In addition, the detailed performance parameters of the system should be introduced, such as the type, frequency, and number of elements of the ultrasonic transducer used. The imaging resolution, field of view, depth, etc. of the system.
We have added these additional details in the methods wherein the Vu-Path device is introduced to better describe the technology.
Round 2
Reviewer 1 Report
Comments and Suggestions for Authors
Dear authors, I don't have further comments or questions.